# Cytotoxicity of Doxorubicin-Curcumin Nanoparticles Conjugated with Two Different Peptides (CKR and EVQ) against FLT3 Protein in Leukemic Stem Cells

**DOI:** 10.3390/polym16172498

**Published:** 2024-09-02

**Authors:** Fah Chueahongthong, Sawitree Chiampanichayakul, Natsima Viriyaadhammaa, Pornngarm Dejkriengkraikul, Siriporn Okonogi, Cory Berkland, Songyot Anuchapreeda

**Affiliations:** 1Department of Medical Technology, Faculty of Associated Medical Sciences, Chiang Mai University, Chiang Mai 50200, Thailand; fah.ch@up.ac.th (F.C.); sawitree.chiampa@cmu.ac.th (S.C.); fai.natsima@gmail.com (N.V.); 2Department of Medical Technology, School of Allied Health Sciences, University of Phayao, Phayao 56000, Thailand; 3Center of Excellence in Pharmaceutical Nanotechnology, Chiang Mai University, Chiang Mai 50200, Thailand; okng2000@gmail.com; 4Cancer Research Unit of Associated Medical Sciences (AMS-CRU), Chiang Mai University, Chiang Mai 50200, Thailand; 5Department of Biochemistry, Faculty of Medicine, Chiang Mai University, Chiang Mai 50200, Thailand; pornngarm.d@cmu.ac.th; 6Department of Pharmaceutical Sciences, Faculty of Pharmacy, Chiang Mai University, Chiang Mai 50200, Thailand; 7Department of Biomedical Engineering and Department of Chemistry, Washington University in St. Louis, Saint Louis, MO 63105, USA

**Keywords:** doxorubicin, curcumin, FLT3, nanoparticle, drug delivery system, peptide, leukemia, stem cell

## Abstract

A targeted micellar formation of doxorubicin (Dox) and curcumin (Cur) was evaluated to enhance the efficacy and reduce the toxicity of these drugs in KG1a leukemic stem cells (LSCs) compared to EoL-1 leukemic cells. Dox-Cur-micelle (DCM) was developed to improve the cell uptake of both compounds in LSCs. Cur-micelle (CM) was produced to compare with DCM. DCM and CM were conjugated with two FLT3 (FMS-like tyrosine kinase)-specific peptides (CKR; C and EVQ; E) to increase drug delivery to KG1a via the FLT3 receptor (AML marker). They were formulated using a film-hydration technique together with a pH-induced self-assembly method. The optimal drug-to-polymer weight ratios for the DCM and CM formulations were 1:40. The weight ratio of Dox and Cur in DCM was 1:9. DCM and CM exhibited a particle size of 20–25 nm with neutral charge and a high %EE. Each micelle exhibited colloidal stability and prolonged drug release. Poloxamer 407 (P407) was modified with terminal azides and conjugated to FLT3-targeting peptides with terminal alkynes. DCM and CM coupled with peptides C, E, and C + E exhibited a higher particle size. Moreover, DCM-C + E and CM-C + E showed the highest toxicity in KG-1a and EoL-1 cells. Using two peptides likely improves the probability of micelles binding to the FLT3 receptor and induces cytotoxicity in leukemic stem cells.

## 1. Introduction

Leukemia is one of the top 10 primary causes of cancer-related mortality, accounting for 3.2% of all cancer cases worldwide [1]. Acute myelogenous leukemia (AML) is a clonal neoplastic growth in the myeloid lineage that is caused by a sequence of somatic mutations in multipotent primitive cells or more differentiated progenitor cells of hematopoietic cells. This buildup of aberrant myeloid cells affects blood circulation and various organs. AML is the most common type of adult acute leukemia and has the highest death rate [2]. Chemotherapy resistance hinders 15–25% of AML patients from achieving complete remission (CR). Furthermore, approximately 50% of patients in CR have a relapse within 5 years.

Leukemic stem cells (LSCs) are one of the main reasons why AML patients relapse and do not respond to treatment [3]. LSCs are derived from the transition of normal hematopoietic stem cells (HSCs), multipotent progenitors (MPPs), or more committed progenitors and can ultimately give rise to leukemia [1]. They can be present in 0.1–1% of 1 × 10^6^ mononuclear cells and are distinguished by the CD34^+^ hematopoietic stem cell surface phenotype and CD38^–-^ subpopulation [4]. LSCs are known to upregulate P-glycoprotein (P-gp). Therefore, it has been identified as one of the mechanisms responsible for the failure of conventional chemotherapy in AML patients [5]. It is important to find new medications or methods for removing LSCs because LSC traits include drug resistance, the ability to self-renew, and propagation of an HSC niche that promotes LSCs.

FLT3 (FMS-like tyrosine kinase 3) or CD135 is a membrane-bound receptor tyrosine kinase protein [6,7]. It is encoded by the *FLT3* gene, which is located on chromosome 13q12 and encompasses 24 exons. More than 90% of AML blasts and most B-cell ALL have high levels of FLT3 expression, and it is less common in chronic leukemia and myeloproliferative disorders [8,9]. Currently, FLT3 is a targeted molecule for AML treatment. Several FLT3 small molecule inhibitors, including midostaurin, lestaurtinib, tandutinib, sunitinib, and sorafenib, have been developed and approved for therapy. In addition to these drugs, the natural compound curcumin has been effective in FLT3-expressing leukemic cells [10]. FLT3 is not only a target for AML treatment, but it also showed great efficacy as a specific receptor for drug delivery systems, as reported by Tima et al. [11].

Doxorubicin or Dox (14-hydroxydaunorubicin; C_27_H_29_NO_11_) is a nonselective class I anthracycline with sugar and aglyconic groups that is approved as an effective treatment for a number of cancers, such as leukemia, Hodgkin’s lymphoma, bladder cancer, and breast cancer [12]. It works by intercalating between DNA base pairs, causing inhibition of DNA and RNA polymerase [13,14]. Anthracyclines constitute the standard treatment for AML. After first-line therapy, it can generate CR in approximately 50–70% of AML patients aged 17–60 [15]. LSCs, however, are reported to be less susceptible to anthracycline [16].

Nanocarrier drug delivery vehicles have been explored to increase the efficacy and decrease the toxicity of Dox. At present, the use of liposomes, polymeric micelles, and nanoparticles has been shown experimentally to improve cancer treatment as carriers of doxorubicin, such as pegylated liposomal doxorubicin (Doxil), polyethylene glycol-pluronic doxorubicin (PEG-pluronic DOX), and N-(2-hydroxypropyl)-methacrylamide copolymer-doxorubicin galactosamine (HPMA-DOX galactosamine; PK2) [17]. Recently, the resolution of multidrug resistance, maximization of treatment efficacy, and reduction of adverse effects have been improved by combining anticancer drugs with another substance engaging a different targeted or chemosensitizing activity [18]. Examples include the combination of daunorubicin and setanaxib (GKT137831), a clinically advanced ROS-modulating substance, venetoclax with hypomethylating agents [19] for AML treatment, or the use of natural substances such as curcumin combined with chemotherapeutic agents such as doxorubicin and cisplatin for cancer treatment [20].

Curcumin (Cur) is a natural polyphenol compound isolated from the rhizome of turmeric (*Curcuma longa* Linn.). It has a cytotoxic effect on AML, CML, and ALL leukemic cell lines, as well as an inhibitory effect on the expression of WT1 and FLT3 proteins and cell proliferation [21,22]. Furthermore, it shows the benefit of alleviating doxorubicin-induced toxicity. It may be used with anthracyclines to diminish anthracycline toxicity and drug resistance in AML leukemic cells and LSCs. However, curcumin has poor in vivo bioavailability because of its exceedingly low water solubility and instability [23]. Liposomes and polymeric micelles, two types of nanocarriers, have been demonstrated to promote bioavailability on AML leukemic cells [24].

In pharmaceutical applications, polymeric micelles are frequently used for the solubilization, stability, and delivery of insoluble drugs [25]. The formation of micelles based on the properties of amphiphilic block copolymers (ABCs) consisting of hydrophilic and hydrophobic blocks that are flexible in number allows them to interact with a variety of biomolecules [25]. Several studies have demonstrated the efficacy of single and combination drug-loaded polymeric micelles in cancer therapy, such as poly(ethylene glycol)-bock-poly(lactide); PEG2k-PLA5k micelles for the co-delivery of Dox and Cur [26], co-encapsulated pegylated polymeric micelles ([DOX + Cur]-PMs) [27] for drug-resistant breast carcinoma therapy, and the combination of Dox-Cur encapsulated biodegradable poly ε-Caprolactone-co-maleic anhydride-graft-citric acid copolymer micelle (PCL-co-P(MA-g-CA)) to eradicate the MDA-MB231 cell line [28].

Poloxamer is an example of an ABC. It is made up of two hydrophilic polyethylene oxide (PEO) blocks and one hydrophobic polypropylene oxide (PPO) block, which are required for micelle formation. Poloxamer 407 (P407) or Pluronic 127 is a non-ionic surfactant with a relative molecular mass of 12.6 kDa (PEO101-PPO56-PEO101). Due to it containing approximately 70% polyoxyethylene groups, this polymer has a high affinity for water [29]. It is utilized in the pharmaceutical industry due to its strong solubilizing capacity, low toxicity, and compatibility with substances and cells. P407 has been used to produce Cur- or Dox-loaded nanoparticles, such as micelles and hydrogels [11,30]. Due to its beneficial properties, P407 was chosen to produce polymeric micelles encapsulating Dox and Cur to improve AML-LSC treatment in this work.

In this study, Dox-Cur-micelles (DCMs) were formulated to increase specificity to targeted cancerous cells by conjugating with two synthetic FLT3-specific peptides (CKR; C and EVQ; E) on the surface of DCM. (DCM-C + E) was developed to improve the efficacy of Dox and Cur delivery to FLT3-positive AML cell lines, especially AML-LSCs. This study demonstrates a novel model of nanoparticles conjugated with two different peptides for drug delivery system improvement. Tima et al. demonstrated that C-CM-micelles and E-CM-micelles could improve the accumulation and cytotoxicity of curcumin in FLT3-positive AML leukemia cells [11]. Moreover, Chueahongthong et al. revealed that Dox-micelles conjugated with both C and E peptides (DM-C + DM-E) exhibited greater cytotoxic activity in KG-1a cells than micelles conjugated with C or E peptides alone [31]. FLT3 ligands (C and E) were synthesized using the sequences CKRFQNSHL (C_48_H_77_N_17_O_13_S_1_) and EVQTCISHLL (C_49_H_83_N_13_O_16_S_1_), respectively. The binding of both peptides to the FLT3 protein receptor close to the high-affinity binding region associated with domains 2 and 3 of the FLT3 extracellular domain [11,24] and this dual targeting of the FLT3 protein on leukemic cell membranes were explored here.

## 2. Materials and Methods

### 2.1. Chemicals and Reagents

Poloxamer 407 (Sigma-Aldrich, St. Louis, MO, USA) (Kolliphor^®^ P 407), dichloromethane (DCM; CH_2_Cl_2_) anhydrous, ≥99.8%, Dess-Martin periodinane, deuterated chloroform (CDCl_3_), tetrahydrofuran (THF; C_4_H_8_O) anhydrous, ≥99.9%, inhibitor-free, dimethyl sulfoxide (DMSO; C_2_H_6_OS), trifluoroacetic acid (TFA; C_2_HF_3_O_2_), (+)-sodium L-ascorbate (C_6_H_7_NaO_6_), crystalline, ≥98%, and aminoguanidine hydrochloride (NH_2_NHC(=NH)NH_2_·HCl), ≥98%, were purchased from Sigma-Aldrich (St. Louis, MO, USA). Ligand THPTA (tris-hydroxypropyltriazolylmethylamine) was purchased from Click Chemistry Tools (Scottsdale, AZ, USA). Azido-PEG6-amine was purchased from BroadPharm (San Diego, CA, USA). FLT3-specific peptides with N-terminal propargylglycine modification, CKR1 (CKRFQNSHL (C_48_H_77_N_1_7O_13_S_1_), MW 1132.29 g/mol), and EVQ (EVQTCISHLL (C_49_H_83_N_13_O_16_S_1_), MW 1237.00 g/mol), were synthesized by Biomatik company (Wilmington, DE, USA). Sodium triacetoxyborohydride (STAB, C_6_H_10_BNaO_6_), 97%, and petroleum ether 40–60 °C, certified AR for analysis were purchased from Fisher Scientific (Pittsburgh, PA, USA). RPMI-1640 Medium, IMDM (Iscove’s Modified Dulbecco’s Medium), L-glutamine, penicillin/streptomycin, dialysis membrane tubing 3500 Dalton MWCO, and phosphate buffered saline (PBS) pH 7.4, 10× solution were purchased from Thermo Fisher Scientific (Waltham, MA, USA). 3-(4,5-Dimethylthiazol-2-yl)-2,5-Diphenyltetrazolium Bromide (MTT) reagent was obtained from BioVision (Milpitas, CA, USA). Amicon^®^ Ultra 15 mL Centrifugal Filters 4500 Dalton MWCO was purchased from MilliporeSigma (Burlington, MA, USA).

### 2.2. Chemotherapeutic Drugs and Curcuminoid Mixture

Doxorubicin-HCl (Dox) (2 mg/mL) for injection (ADRIM, Fresenius Kabi, Pune, India) and Doxorubicin Hydrochloride Salt were purchased from Maharai Nakhorn Chiang Mai hospital (Chiang Mai, Thailand) and LC Laboratories (Woburn, MA, USA), respectively. Curcumin powder was purchased from Thai-China Flavours and Fragrances Industry Co., Ltd. (Nonthaburi, Thailand). Cur solution was prepared at a concentration of 25 µg/mL in DMSO as a stock solution and kept at −20 °C in the dark.

### 2.3. Leukemic Cell Lines and Cell Culture Conditions

The four AML leukemic cell lines, including KG-1a (RBC1928, RIKEN BRC, Tsukuba, Japan) and KG-1 cells (ATCC^®^ CCL-246^TM^) (model of FLT3-overexpressing AML stem cell-like-leukemic cells), EoL-1 cells (RCB0641, RIKEN BRC, Japan) (model of FLT3-overexpressing AML leukemic cells), and U937 cells (generous gift from Prof. Dr. Watchara Kasinrek) (model of non-FLT3-expressing AML leukemic cells), were used as the myeloid cell line models in this study.

### 2.4. Preparation and Physicochemical Characterization of Drug-Loaded Micelles

#### 2.4.1. Method for DCM and CM Preparation

DCM and CM were produced by combining evaporation-induced self-assembly and pH-induced self-assembly techniques.

P407 and Cur were dissolved in methanol at concentrations of 10 and 1 mg/mL, respectively, to produce DCM. Cur-micelle-film was created by evaporation in a rotary evaporator at 45 °C, 50 rpm for 20 min, followed by overnight drying at RT. The micelle-film was then reconstituted in normal saline solution (NSS) and agitated for 30 min at 350 rpm, before 10× PBS, pH 7.4, and Dox solution were added during agitating to obtain DCM after an overnight incubation. At the end of the process, the volumes of NSS, 10× PBS, and Dox solution were 7:1:2. After 10 min of centrifugation at 12,000 rpm, the DCM was filtered through a 0.22 μm membrane to eliminate any unbound drug. Clear DCM solution was collected and kept in a light-resistant container. Similar to the DCM preparation, P407 and Cur were dissolved in methanol for the CM preparation. The Cur-micelle-film was then produced by evaporation and rehydrated in NSS and PBS, pH 7.4, respectively.

#### 2.4.2. Optimization Conditions for DCM and CM Preparation

In order to obtain the most suitable formulation for preparing DCM and CM by the solvent evaporation and pH-sensitive method, the amounts of polymer, Dox, and Cur used for the DCM and CM preparations were varied in many conditions.

The weight ratio of Cur:P407 (mg) employed for CM optimization was 1:20 (CM1), 1:24 (CM2), 1:30 (CM3), 1:40 (CM4), 1:50 (CM5), 1:60 (CM6), and 1:80 (CM7). For the DCM optimization conditions, the weight ratio of Dox-Cur:P407 (mg) was 1:20 (DCM1), 1:30 (DCM2), 1:40 (DCM3), 1:50 (DCM4), 1:60 (DCM5), 1:70 (DCM6), and 1:80 (DCM7).

For the DCM formulation, KG-1a, KG-1, and EoL-1 cells were incubated in a Dox-Cur solution having the maximum concentration of Dox-2 µg/mL. In each condition, the highest concentrations of Dox to Cur were Dox-2 µg/mL to Cur-18 µg/mL (1:9), Cur-20 µg/mL (1:10), Cur-28 µg/mL (1:14), Cur-38 µg/mL (1:19), Cur-48 µg/mL (1:24), and Cur-58 µg/mL (1:29). The maximum concentration of Dox used in U937 cells was 0.5 µg/mL. The highest concentrations of Dox to Cur administered in each condition of Dox-Cur solution were Dox-0.5 µg/mL to Cur-4.5 µg/mL (1:9), Cur-5 µg/mL (1:10), Cur-7 µg/mL (1:14), Cur-9.5 µg/mL (1:19), Cur-12 µg/mL (1:24), and Cur-14.5 µg/mL (1:29). Cell viability was determined using the MTT assay. In this study, the weight ratio of Dox and Cur was fixed at 1:9 in every condition of the DCM.

In brief, KG-1a (1.5 × 10^4^ cells/100 µL), KG-1 (1.5 × 10^4^ cells/100 µL), EoL-1 (3.0 × 10^4^ cells/100 µL), and U937 (1.0 × 10^4^ cells/100 µL) cells were seeded in a 96-well plate and incubated at 37 °C, 5% CO_2_ overnight. The cells were cultured for 48 h with 100 µL of each treatment with various Dox-Cur concentration ratios. Complete medium and DMSO were used as the cell control (CC) and vehicle control (VC), respectively. After removing 100 µL of medium, 15 µL of MTT dye solution was added, followed by a 4 h incubation. To dissolve the formazan crystals, the entire solution was discarded, and DMSO (200 µL) was added to each well. The optical density (OD) of formazan solution was examined at 578 nm and 630 nm as the reference wavelength by using a microplate reader. The percentage of surviving cells was determined using the following equation.
(1)$$\%\;\mathrm{Cell}\;\mathrm{viability}\;=\frac{\mathrm{Mean}\;\mathrm{absorbance}\;\mathrm{in}\;\mathrm{test}\;\mathrm{well}}{\mathrm{Mean}\;\mathrm{absorbance}\;\mathrm{in}\;\mathrm{vehicle}\;\mathrm{control}\;\mathrm{well}}\times 100$$

#### 2.4.3. Characterization of DCM and CM Physicochemical Properties

The dynamic light scattering (DLS) approach was used to evaluate the particle size (PZ), polydispersity index (PdI), and zeta potential (ZP) (Particle size analyzer, Brookhaven Instruments Corporation, Holtsville, NY, USA). The micelle morphology was examined using a Transmission Electron Microscope (TEM) (Hitachi High-Technologies Corporation, Tokyo, Japan). To determine the Dox and Cur concentrations in DCM and CM, the samples were diluted with methanol in a volume ratio of 1:1 and centrifuged at 5000 rpm for 5 min to break the micelles. Dox and Cur in DCM were then determined using an RP-HPLC (Waters Alliance 2795 HPLC with 2489 UV/Visible Detector, Waters Corporation, Milford, MA, USA) equipped with a C18 column (5 µm and 4.6 × 250 mm). The linear gradient condition for detecting Dox and Cur consisted of 5 to 95% acetonitrile in water with constant 0.05% trifluoroacetic acid, a flow rate of 1.0 mL/min, a detection wavelength of 280 nm, and a 20 min run time. Cur in CM was measured by HPLC or a UV–visible spectrophotometer at 425 nm.

The loading capacity (%LC) and entrapment efficiency (%EE) of each component in the micelles were calculated using the Dox and Cur contents in accordance with the Dox and Cur standard curves in methanol as follows:(2)$$\%\;\mathrm{LC}\;=\frac{\mathrm{Amount}\;\mathrm{of}\;\mathrm{doxorubicin}\;\mathrm{in}\;\mathrm{nanoparticles}}{\mathrm{Amount}\;\mathrm{of}\;\mathrm{nanoparticles}}\times 100$$
(3)$$\%\;\mathrm{EE}\;=\frac{\mathrm{Measured}\;\mathrm{amount}\;\mathrm{of}\;\mathrm{doxorubicin}\;\mathrm{in}\;\mathrm{nanoparticle}\;}{\mathrm{Total}\;\mathrm{amount}\;\mathrm{of}\;\mathrm{doxorubicin}\;\mathrm{used}}\times 100\;$$

#### 2.4.4. Study of the Interaction between Dox, Cur, and P407 in DCM and CM Using XRD and DSC Analyses

To determine the status of Dox and Cur encapsulated in DCM and CM, the crystalline and thermal properties of each micelle were measured. DCM and CM solutions were frozen at −80 °C for at least 6 h and then lyophilized to collect the micelles in powder form. The crystallinity of the lyophilized micelles, Dox powder, Cur powder, and P407 was examined using an X-ray diffractometer (Rigaku, Tokyo, Japan) with the scattering angle 2θ from 0–60° at room temperature. Origin software version 8.5 was utilized for data analysis.

The melting point of the lyophilized micelles, Dox powder, Cur powder, and P407 (3–5 mg/sample) was investigated by using a differential scanning calorimeter (DSC) at various temperatures ranging from 0–300 °C with a heating rate of 5 °C/min (TA Instruments, New Castle, DE, USA). The data were analyzed using version 4.5A of the Universal Analysis 2000 software.

#### 2.4.5. In Vitro Dox and Cur Release Profile in DCM and CM

The accumulative percentage of Dox and Cur released from DCM and CM was evaluated by using the dialysis method. A dialysis tube with a MWCO of 3500 Da was filled with drug-micelles, single drug solution, and combined drug solution (3 mL total). The tubes were then shaken continuously at 37 °C, 100 rpm, for 72 h in 30 mL of PBS, pH 7.4, containing 0.5% *w*/*w* Tween 80, in a container that was resistant to light. At 0, 1, 2, 4, 6, 8, 10, 12, 24, 48, and 72 h, release medium (1 mL) was collected, and the same volume of new medium was introduced to replace the withdrawal. Using HPLC, the amount of Dox and Cur in each time-collected sample was determined, and the concentration of each compound was estimated using the standard curve.

### 2.5. Production, Purification, and Assessment of P407-FLT3 Peptide Conjugation for the Drug-Micelle Conjugated FLT3 Peptide Preparation

#### 2.5.1. Production and Purification of P407-FLT3 Peptide Conjugation

P407-CHO and CA production

The terminal hydroxy group (–OH) of P407 was modified to an aldehyde group (–CHO), and the resulting product was coupled with an amino-PEG azide linker to generate the cargo azide (CA), yielding P407 with the functional azide required for peptide conjugation as reported in our previous study [31].

CA-conjugated FLT3-peptide preparation

CA conjugated with CKR (CA-CKR), CA conjugated with EVQ (CA-EVQ), and CA conjugated with CKR and EVQ (CA-CKR-EVQ) were synthesized by conjugating CA to alkyne-modified peptides, CKR (MW 1132.29 g/mol) and EVQ (1237.00 g/mol), via a copper-catalyzed azide-alkyne (CuAAC) reaction as described in previous studies [31,32]. Until usage, all modified polymers were kept at −20 °C. In this experiment, potassium phosphate (KH_2_PO_4_) buffer was utilized as a solvent for the preparation of alkyne-peptide and CA solution.

#### 2.5.2. The Evaluation of P407-FLT3 Peptide Conjugation

To confirm the success of the production of P407-CHO, CA, and CA-peptides, each sample was dissolved in chloroform-d (CDCl_3_) at a concentration of 20–30 mg/600 µL, and the changes in ^1^H-NMR and HPLC were investigated.

For the analysis of CA-peptide conjugation using HPLC, the reaction was separated under two conditions: pre-peptide-ASC (alkyne-peptide, cargo-azide, CuSO_4_, and THPTA) and complete-peptide-ASC (alkyne-peptide, cargo-azide, CuSO_4_, THPTA, and Na-ascorbate (ASC)). The peak of the CKR and EVQ peptides was identified at 214 nm using a linear gradient with 5–95% acetonitrile in water with constant 0.05% trifluoroacetic acid at 1 mL/min for 20 min. If the peak of each peptide in the complete reaction was lower than the peak of the peptide in the reaction without ASC, the peptide could couple with the cargo-azide.

Furthermore, a scanning UV-Vis Spectrophotometer (UV2600i, Shimadzu, Kyoto, Japan) was used to analyze the UV-Vis absorption spectra of P407, CA, and CA-peptide conjugates between 200 and 800 nm.

### 2.6. DCM and CM Conjugated with FLT3 Peptide Preparation and Physicochemical Assessment

DCM and CM formulations were prepared with a weight ratio of drug to individual and two modified polymers (CA-CKR, CA-EVQ, and the mixture of CA-CKR and CA-EVQ) of 1:40 mg and a weight ratio of Dox to Cur of 1:9 for DCM. DCM conjugated with FLT3-specific peptides included DCM-CKR (DCM-C), DCM-EVQ (DCM-E), and DCM-CKR + EVQ (DCM-C + E). CM conjugated with FLT3-specific peptides included CM-CKR (CM-C), CM-EVQ (CM-E), and CM-CKR + EVQ (CM-C + E).

The modified polymers (40 mg) and Cur (0.9 mg for DCM and 1 mg for CM) were dissolved in methanol at a concentration of 1 mg/mL and then evaporated and dried at room temperature as described in the previous section. The CM-films were rehydrated in 1.4 mL of NSS, followed by 0.2 mL of 10× PBS and 0.4 mL of Dox solution (0.1 mg of Dox) or NSS while stirring to produce DCM and CM conjugated with FLT3 peptides.

The particle size, PdI, zeta potential, %EE, and %LC of DCM and CM conjugated with FLT3 peptides were measured.

### 2.7. Evaluation of Cell Viability after Treatment with Dox Solution, Dox-Cur Solution, and DCM with or without FLT3 Peptides by MTT Assay

KG-1a (1.5 × 10^4^ cells/100 µL) and EoL-1 (3 × 10^4^ cells/µL) cells were seeded in flat-bottom 96-well plates for 24 h. Then, 100 µL of medium, DMSO (vehicle control), Dox-HCl, Dox-Cur solution (Dox 4 µg/mL + Cur 36 µg/mL; ratio 1:9), DCM, DCM-CKR, DCM-EVQ, and DCM-CKR + DCM-EVQ at concentrations of Dox ranging from 0–4 µg/mL (2-fold dilution) were added and incubated for 48 h (the final concentrations of Dox were 0–2 µg/mL). As described in Section 2.4.2, the treated cells were then incubated with MTT solution, and the optical density (OD) of each sample was determined. To assess the toxicity of the drug in micelle and solution forms, the percentage of viable cells was plotted as a dose-response curve, and the IC_50_ value was obtained.

### 2.8. Statistical Analysis

All results are shown as the mean and standard deviation (SD) or mean and standard error of the mean (SEM). The statistical significance of differences between the control, drug solution, and various drug-micelle formulations was assessed using a one-way ANOVA. The data were statistically significant when the *p*-value < 0.05.

## 3. Results

### 3.1. Preparation and Evaluation of the Physicochemical Properties of Drug-Micelles

#### 3.1.1. Formula Optimization and Characterization of DCM and CM

To formulate DCM, the optimal concentration ratio of Dox and Cur was investigated by the MTT assay in KG-1a, KG-1, EoL-1, and U937 cells. Thus, various co-treatment conditions of Dox and Cur solutions, including Dox and Cur at concentration ratios of 1:9, 1:10, 1:14, 1:19, 1:24, and 1:29, were incubated with all AML cell models to examine the suitable ratio of Dox to Cur for DCM formulation.

The lowest Dox-Cur ratio that might affect KG-1a and KG-1 leukemic stem cell cytotoxicity was found to be 1:9 and 1:14, respectively (Figure 1A,B). Compared to other cell lines, KG-1a had the highest number of leukemic stem cell populations, which were the target of this study. Therefore, a co-treatment ratio of 1:9 was utilized for Dox-Cur micelle production. However, in EoL-1 and U937 leukemic cells, no significant difference was observed between samples treated with Dox-Cur co-treatment and those treated with Dox alone, most likely due to the low concentration of Cur used in the co-treatment (Figure 1C,D).

After selecting the concentration ratio of Dox and Cur of 1:9, the various weights of Dox-Cur and P407 were used to prepared DCM by using the film-hydration method in conjunction with a pH-induced self-assembly method. The results demonstrated that the mean particle size, PdI, and zeta potential of freshly prepared micelles were 21–27 nm, 0.148–0.162, and −2 to −5 mV, respectively (Appendix A). For the zeta potential, all DMs had nearly neutral surface charge (−10 to 10 mV). After 5 days of RT storage, all conditions of DCM also presented a fine particle size (21–29 nm) with a narrow size distribution (PdI < 0.300). However, the %EE of Dox and Cur in DCM3 was higher than 80 and 90% of initial loading drugs (Appendix A). Therefore, DCM with a drug and polymer weight ratio at 1:40 (DCM3), which had a suitable particle size along with a high %EE and %LC, was chosen for DCM preparation.

For CM, the weight ratio of Cur to P407 was adjusted from 1:20 to 1:80 to prepare various CM formulations via the same method used for DCM production. However, CM was completely formed only when using the film hydration technique and when 10× PBS, pH 7.4, was added to achieve the same conditions as DCM. From the physicochemical properties, all formulations of freshly prepared CM had particle sizes in the range of 23–28 nm with a PdI value < 0.300, and the zeta potential was −3 to 0, which was near neutral as shown in DCM (Appendix A). For the Cur content, the %EE of Cur in CM3, CM4, and CM7 was found to be higher than 80%, indicating that Cur was well-encapsulated in CM. CM4 and CM7 maintained the particle size at nearly that of the fresh one after 5 days of RT storage. However, the %LC of CM4 was higher than CM7. As indicated in Appendix A, the CM4 formulation with a weight ratio of Cur to P407 of 1:40 mg was determined to be the optimal formulation for CM synthesis.

For blank micelle (BM), the particle size, PdI, and zeta potential were 25.78 ± 2.11 nm, 0.107 ± 0.010, and −0.44 ± 0.63 mV, respectively, showing that Dox-Cur or Cur encapsulation in micelles did not affect the particle size and charge of micelles. For physical appearance, DCM and CM were clear orange and yellow solutions, respectively, indicating that Dox-Cur and Cur could be well-encapsulated in the core-shell micelles (Figure 2). Furthermore, the morphology of DCM and CM was determined using TEM. Corresponding with the results of the Zetasizer-based dynamic light scattering technique, a spherical nanoparticle with a size below 100 nm was observed (Figure 2).

#### 3.1.2. The Interaction between Dox-HCl, Cur, and P407 in DM, CM, and DCM

The entrapment of Dox-Cur and Cur only in poloxamer micelles was determined using XRD and DSC analyses. According to XRD, Dox and Cur crystalline structures displayed prominent peaks at scattering angles of 13° to 25° and 15° to 30°, respectively. Meanwhile, the diffraction peaks of P407 and BM were detected at 19° and 23°, respectively. The typical peak of P407 was only observed in DCM and CM samples, suggesting that Dox and Cur were entrapped within the micelles in an amorphous state. In addition, a new peak around 27° was found in BM, DCM, and CM samples, indicating that the micelle formation process could be influenced by a novel fine arrangement of P407 molecules. Notably, the diffraction peaks of NaCl in PBS were observed at 33°, 46°, and 55° because 10× PBS, pH 7.4, was used in the micelle preparation step. The XRD patterns of P407, Dox, Cur, lyophilized BM, lyophilized DCM, lyophilized CM, and NaCl are presented in Figure 3A,B.

In order to confirm drug encapsulation, the thermal behavior of DCM and CM was examined using DSC analysis. The DSC thermograms showed an endothermic melting peak of P407 at 56.67 °C; Dox at 204.03, 225.99, and 236.96 °C; and Cur at 177.88 °C. The thermograms of DCM and CM exhibited the same endothermic peaks of P407 as those of P407 and BM; however, the melting peaks of Dox and Cur were absent from these samples, suggesting the absence of crystalline Dox and Cur within the micelle. In addition, the results from the mixed powders of P407-Dox-Cur and P407-Cur showed all the melting peaks of each component in the mixed samples, confirming these findings. In P407 micelles, however, there was little change in the melting point and the existence of a broad peak (80–90 °C) that could be involved in the new crystalline structure formation. Figure 4A,B shows the thermal behavior of P407, Dox, Cur, mixed powders, lyophilized BM, lyophilized DCM, and lyophilized CM.

#### 3.1.3. Effects of Colloidal, Temperature, and Time on DCM and CM Stability

To assess the colloidal stability, DCM and CM were incubated in PBS solution containing various concentrations of BSA solution, and the alterations in the physicochemical properties were determined (Figure 5 and Figure 6). After 24 h of incubation time, the mean particle size of DCM dissolved in PBS, pH 7.4, with 0, 1, 5, and 10% BSA was 22.87 ± 0.05, 18.79 ± 0.10, 26.49 ± 0.87, and 41.58 ± 5.05 nm, respectively (Figure 5A), while the mean particle size of CM was 22.89 ± 0.27, 21.32 ± 0.76, 34.75 ± 2.82, and 67.18 ± 5.05 nm, respectively (Figure 6A). The PdI values slightly decreased (Figure 5B and Figure 6B). The significant increase in the size of DCM and CM in 10% BSA in PBS solution suggested that a high concentration of BSA might contribute to the aggregation of micelles. In addition, the zeta potential of all the micelles in samples with a high content of BSA was considerably more negative, although they remained nearly neutral charged (Figure 5C and Figure 6C). Despite the increasing BSA concentration, the particle size and zeta potential of DCM and CM in each PBS solution containing BSA were comparable. Therefore, under physiological conditions, the circulation of these micelles may be prolonged.

Furthermore, the stability of DCM and CM under RT, 4, and −80 °C was evaluated. The changes in the physicochemical properties of the micelles in each storage condition were determined within 1 month at room temperature. The results showed that the physical properties of DCM and CM stored at −80 °C were still comparable to fresh samples (particle size < 30 nm). On the other hand, relative to their initial sizes, DCM and CM grew slightly in size at room temperature and 4 °C storage (<50 nm), compared to the initial size. It appeared that temperature might increase the probability of micelle aggregation. Long-term storage could impact the PdI and zeta potential of micelles, particularly those stored at room temperature and 4 °C, but no significant changes were detected.

Additionally, the entrapment efficiency of Cur in CM maintained at −80 °C was approximately greater than 90%, and similarly to DCM, the Dox and Cur contents could be preserved at 70 and 90%, respectively, which was comparable to freshly prepared samples after 30 days of storage. The Dox content gradually decreased more than Cur during storage at 4 °C and room temperature. In contrast, the decline in the Dox and Cur contents in samples stored at 4 °C was slower than at room temperature, showing that low temperature was appropriate for preserving drug-encapsulated micelle solutions. These results were consistent with the %LC that remained constant when the samples were stored at −80 °C. Therefore, storage duration and temperature could have an effect on the quality of DCM and CM, with −80 °C being the most effective temperature for sustaining the physicochemical stability of these micelles in solution form. The physicochemical changes in DCM and CM are shown in Figure 7 and Figure 8.

#### 3.1.4. In Vitro Release Profile of DCM and CM

DCM and CM were evaluated for drug release in PBS at pH 7.4, compared to drug in soluble form using the dialysis method. The cumulative release profiles of Dox in Dox-Cur solution and DCM at 24 h were 65.30 ± 2.12 and 56.26 ± 3.59%, respectively. For Cur in Dox-Cur solution and DCM, the values were 56.98 ± 3.17 and 2.37 ± 0.26%, respectively. Similar to the cumulative release profiles at 24 h, Cur solution and CM were 51.58 ± 3.54 and 4.05 ± 0.41%, respectively. The release of Dox and Cur in Dox-Cur and Cur solution was found to be significantly higher than the release of Dox and Cur from DCM (Figure 9A) and CM (Figure 9B). These data suggested that P407 polymeric micelles had the ability to encapsulate Dox and Cur within the hydrophobic core of the micelles, hence extending the drug release rate. However, the release of Dox was apparently higher than Cur due to the stronger hydrophobicity of Cur than Dox, which resulted in stronger encapsulation in hydrophobic inner core of micelles of Cur. These findings might be related to the fast decline in Dox in DCM compared to Cur in DCM and CM stored at room temperature and 4 °C. The cumulative release profiles of Dox-Cur and Cur in solution and micelle form are shown in Figure 9A,B.

### 3.2. Synthesis and Characterization of P407-FLT3-Specific Peptide Conjugates

P407 was modified and coupled with FLT3 peptides in order to produce micelle-conjugated single and two different FLT3 peptides. P407 was first modified to P407-CHO using DMP, which could oxidize primary alcohols to aldehydes. The complete reaction was confirmed by the appearance of the CHO-group peak at δ = 9.75 ppm in the ^1^H-NMR spectrum. After that, P407 with a functional azide group, cargo azide (CA), was formulated by coupling P407-CHO with an azido-PEG6-amine linker. The disappearance of a CHO-peak in ^1^H-NMR suggested that P407 and the amine linker were properly conjugated. The NMR spectra of P407-OH, P407-CHO, and cargo-azide are shown in Chueahongthong et al. [31].

The CA and specific alkyne-modified FLT3-specific peptides CKR and EVQ were used to synthesize polymer conjugated with single or two FLT3 peptides, including CA-C, CA-E, and CA-C-E, through the CuAAC reaction. In the ^1^H-NMR spectrum of CA-C, CA-E, and CA-C-E, the results indicated the presence of a novel triazole ring peak at 8.02–8.12 ppm (Figure 10). However, the NMR patterns of the three CA-conjugated FLT3 peptides were similar. The success of the conjugation of CA-C, CA-E, and CA-C-E was determined using HPLC.

For HPLC detection, samples were prepared in two vials, one containing alkyne-peptide, cargo-azide, CuSO_4_, and THPTA (pre-peptide-ASC), and the other containing alkyne-peptide, cargo-azide, CuSO_4_, and ascorbic acid (ASC) (peptide-ASC). CKR and EVQ peptides were identified at 214 nm. The decrease in the area under the curve (AUC) of the CKR and EVQ peptides peaks in the complete reaction vial compared to the peptide peak in the pre-ASC vial suggested that the peptide could bind to cargo-azide.

C (8.663 min) and E (9.845 min) peptides after reaction had longer retention times than the peptide peak in pre-ASC and standard vial in all CA conjugated with one individual or both FLT3 peptides, showing that both peptides were able to bind cargo-azide in a one- and two-peptide conjugation reaction. Unfortunately, it was observed that CKR and EVQ could form disulfide bonds, which could affect the concentration of both peptides during the conjugation process (Figure 11A–C). Instead of CA-C-E, CA-C and CA-E were used to prepare micelles conjugated with two distinct FLT3 peptides.

In addition, scanning UV-Vis spectroscopy was utilized to validate the success of the CA-peptide conjugation process. The appearance of a strong absorption peak of peptides at 210 nm in the conjugation samples, similar to unconjugated peptides and in contrast to CA and P407 samples, supported the results (Figure 12).

### 3.3. Preparation and Characterization of DCM and CM Conjugated with C, E, and Both C and E Peptides

After polymer–peptide conjugation, DCM and CM conjugated with two FLT3 peptides were produced by using the film-hydration and pH-induced self-assembly method at a weight ratio of drug to polymer at 1:40 as mentioned in the drug-micelles without peptide preparation.

Clear orange and yellow solutions of DCM (Figure 13A) and CM (Figure 13B) formulations conjugated with FLT3 peptides were collected after processing. According to the data in Appendix A, all newly synthesized DCM and CM with C, E, and C + E demonstrated a narrow particle size distribution between 25 and 45 nm and a low zeta potential between −1 and −5 mV; nevertheless, the mean particle size of DCM and CM coupled with FLT3 peptides was larger than that of those without peptides. The Dox and Cur contents in micelles coupled with one or two peptides were also lower than in P407 without peptides. The %EE of Dox and Cur in DCM and DCM conjugated peptides was 80–90% and 70–90%, respectively, which was similar to the %EE of Cur in CM conjugated with peptides, which was roughly 70–90%. However, the drug content in the micelles was greater than 70%, showing that Dox and Cur could be packaged in the core-shell micelles. According to TEM examination, each formulation of micelles conjugated with peptides contained spherical nanoparticles smaller than 100 nm, which was consistent with the size distribution histogram data (Figure 14).

### 3.4. Determination of Cytotoxic Effects of Dox Solution, Dox-Cur Solution, and DCM Conjugated with or without FLT3 Peptides on Leukemic Cell Viability by MTT Assay

KG-1a and EoL-1 cells were treated with Dox, Dox-Cur solution, and DCM conjugated with and without peptides at equivalent Dox doses of 0–2 μg/mL (the concentration ratio of Dox to Cur was 1:9) for 48 h to evaluate the cytotoxicity of co-treatment of Dox-Cur and DCM formulations. The KG-1a cell viability curve and IC_50_ values of Dox revealed that the Dox-Cur solution and all formulations of DCM had a stronger cytotoxic effect than Dox solution by evaluating the concentration of Dox in each sample. Comparing DCM with or without peptides and the Dox-Cur solution did not exhibit significant differences. These results might be attributable to the concentrations of Dox and Cur in DCM formulations, since the concentration ratio of Dox and Cur encapsulated in the micelles was difficult to control. Therefore, it likely reduced the cytotoxicity of the DCM formulation in KG-1a cells. However, DCM-C + E exhibited the highest toxicity compared to other conditions (Figure 15A and Table 1).

There was no difference in cytotoxicity between Dox solution, Dox-Cur solution at a concentration ratio of 1:9, and all DCM formulations for EoL-1 cells (Figure 15B and Table 1), which might have come from the Dox sensitivity of EoL-1 cells and the Dox-Cur ratio issue in the DCM formulations. To investigate the cytotoxicity of co-treatment with Dox and Cur in micelle form, the Dox-micelle (DM) formulations described in our earlier work [33] were combined with CM formulations.

### 3.5. Determination of Cytotoxic Effects of Cur Solution and CM Conjugated with or without FLT3 Peptides on Leukemic Cell Viability by MTT Assay

For the cytotoxic assessment of Cur solution and CM formulations, KG-1a and EoL-1 cells were treated with Cur solution, CM, CM-C, CM-E, and CM-C + CM-E at equivalent Cur concentrations of 0–25 µg/mL for 48 h. The results showed that only CM-C and CM-C + CM-E exhibited higher toxicity than Cur solution and CM in KG-1a cells (Figure 15C and Table 2). On the other hand, the cytotoxicity of all CM coupled with peptides was significantly greater than that of Cur solution and CM in EoL-1 cells (Figure 15D and Table 2). However, CM demonstrated a significantly lower toxic effect than the Cur solution in both KG-1a and EoL-1 cells; this was probably because Cur solution was dissolved in DMSO, rendering it highly soluble in the medium and capable of passing directly through the cell membrane. The IC_50_ values of Cur solution and CM with or without FLT3 peptides in KG-1a and EoL-1 cells are shown in Table 2.

## 4. Discussion

LSCs are regarded to be one of the primary causes of relapse in acute myeloid leukemia (AML) patients due to their resistance to standard chemotherapy drugs, such as Dox. The combination of Dox and Cur, a natural polyphenol with known anticancer and chemosensitizing properties, was explored in this study to target LSCs more effectively while minimizing the side effects associated with Dox. Previous research demonstrated that Cur could enhance the cytotoxic effects of Dox against AML stem cells (KG-1a and KG-1 cells) and leukemic cells (EoL-1 and U937 cells) [31].

To improve the permeability, retention, and pharmacokinetic properties of Dox and Cur while reducing adverse effects, a specific drug delivery system based on nanocarriers has been developed to more selectively deliver drugs to tumors. Several recent studies have demonstrated the effectiveness of using nanocarriers combined with specific peptides to increase the delivery of Dox and Cur in different types of cancer. For example, the targeted delivery of Dox and Cur to breast and lung cancer was achieved by combining the Arginine-Glycine-Aspartic acid (RGD) peptide, which can bind to integrin, with graphene quantum dots (GQDs) [34] or MPEG-PLA polymer [35], respectively. Moreover, Dox and Cur encapsulated in micelles formed from the amphiphilic small molecule peptide Pep1 (DC/Pep1) was developed to inhibit the growth and metastasis of hepatocellular carcinoma [36]. However, few studies have focused on developing the delivery of Dox and Cur using nanoparticles conjugated with cell-specific peptides, particularly in AML. To date, there have been no reports of combined drug-loaded nanoparticles that improve specificity to targeted cancer cells by using multiple types of peptides [37,38]. This study presents a novel approach to increase Dox and Cur cytotoxicity in AML cell lines by demonstrating the development and utilization of Dox-Cur micelles conjugated with two FLT3 peptides, which can bind to different sites on the FLT3 receptor in AML LSCs and leukemic cells. To produce the micelles in this study, the triblock co-polymer P407 was chosen [25,39,40,41]. P407 was further modified and conjugated with two FLT3-specific peptides (C and E), which were synthesized by Tima et al. to increase Dox and Cur delivery to LSCs and increase its safety at relatively large cumulative dosages [11]. This is the first report to design a novel strategy model for drug delivery systems.

The film-hydration and pH-induced self-assembly methods were used to synthesize DCM and CM [30]. Cur and P407 were loaded into micelles via self-assembly and were readily encapsulated into the core-shell structure of P407 micelles. Dox was observed to precipitate in phosphate buffer saline (PBS) solution at pH 7.4 and under these conditions could be entrapped with Cur to produce DCM [30,42]. A suitable weight ratio of drug to polymer of DCM and CM was 1:40, and the particle size ranged between 20 and 25 nm. The literature indicates that micelles must be sufficiently smaller than 200 nm to effectively permeate tissue and evade the clearance by renal filtration and the reticuloendothelial system (RES) [43,44]. We also observed a micelle surface charge from 0 to −5 mV, which is close to neutral (−10 to 10 mV [45]). Numerous micelle reports [27,30,46,47] have indicated that a low zeta potential can cause particle agglomeration due to van der Waals forces [16]; however, the hydrophilic PEO block of P407 has the capability of preventing aggregation, protein absorption, and recognition of the particle by the RES [25,46]. Plasma protein adsorption on an intravenous drug carrier is also a factor that dictates the in vivo behavior of nanoparticles [48]. We found DCM and CM maintained a size comparable to that incubated in PBS, pH 7.4, although incubation in highly concentrated BSA (10%) increased the average particle size and zeta potential. Finally, the %EE of DCM and CM was greater than 80% as amorphous Dox and Cur, as evidenced by XRD and DSC analyses.

To study the efficacy of drug uptake by AML-LSCs, the C and E peptides targeting FLT3 were conjugated to P407 using a “click” (CuAAC) reaction. The C and E peptides have been demonstrated to bind different regions of the FLT3 receptor [11]. Our previous study demonstrated that Dox micelles coupled with C and E peptides might improve Dox uptake, cytotoxicity, and apoptosis in KG-1a cells [31]. Here, we were interested in exploring the possible synergy of the C and E peptides.

The size, zeta potential, and drug content of DCM and CM conjugated with C, E, or C + E did not differ in terms of their physicochemical properties. The molecular weight of C (1132.29 g/mol) is similar to that of E (1237.00 g/mol), and the same concentration ratios of peptide and CA (1:2) were used for preparing polymer–peptide conjugates. We observed that the physicochemical characteristics of micelles displaying C, E, or C + E were similar. In comparison, DCM and CM without peptides exhibited a smaller particle size and higher drug content than the conjugation group [49].

No differences were observed between DCM formulations (micelles) when compared to Dox-Cur solutions. This may be the result of differences in the Dox and Cur ratio in DCM. The concentration ratio of Dox to Cur in DCM with or without peptides varied from 1:7 to 1:10. Because the P407 micelles containing a single drug were unable to adequately encapsulate the entire drug load, variations in the Dox and Cur amounts were observed in DCM. Furthermore, DCMs displaying FLT3 peptides also showed some variation in drug encapsulation. Therefore, small differences in the content ratios of Dox and Cur in each formulation might have reduced the cytotoxicity of the DCM formulation on KG-1a and EoL-1 cell models. Reports suggest that low drug loading in nanocarriers may not completely overcome multidrug resistance caused by exposure to free drugs [50]. Even so, DCM-C + E tended to show the highest cytotoxicity in both KG-1a and EoL-1 cells, compared to other combined drug formulations. Thus, DCM with improved drug encapsulation may improve the efficacy for LSC treatment. From our study of the synergistic effects of Dox-Cur solution in EoL-1 cells, the concentration of Cur should be higher than Dox by about 100 times to achieve drug synergy. Specifically, we also observed that the concentration ratio of Dox-Cur at 1:9 was only appropriate for KG-1a cells. For the study of the cytotoxicity of CM formulations, CM coupled FLT3 peptides had lower IC_50_ values than CM and Cur solution, and CM-C + E exhibited the lowest IC_50_ value in both KG-1a and EoL-1 cells, suggesting that FLT3 peptides could enhance the cytotoxicity of CM in leukemic stem cells. This finding was consistent with the previous study of Tima et. al. [11]; however, further studies are needed to discern any benefit of using two FLT3 targeting peptides on nanocarriers [39]. From this study, the co-delivery of drugs in polymeric micelles conjugated with two peptides is expected to improve potential clinical outcomes for AML patients in the future.

## 5. Conclusions

In the present study, DCM and CM were successfully synthesized using a drug-to-poloxamer 407 weight ratio of 1:40. All micelles displayed nanoscale particle size, efficient drug encapsulation, and physicochemical stability at −80 °C for 30 days. In addition, DCM and CM exhibited sustained drug release and stable micelle size and zeta potential when incubated in a mock physiological environment. Micelles displaying FLT3 peptides, C and E, showed similar physicochemical properties to those without peptides. The cytotoxic effect of DCM with or without FLT3 peptides against LSCs did not differ from that of their respective drug solutions. However, CM conjugated with FLT3 peptides exhibited significantly greater cytotoxicity against KG-1a stem cells and EoL-1 leukemic cells compared to Cur solution. Our findings indicated that P407 micelles conjugated with FLT3 peptides increased the cellular uptake of Dox and Cur, leading to an increase in drug accumulation in the cells, efficiently eliminating LSCs. In conclusion, DCM and CM conjugated with FLT3 peptides offer a more selective AML-LSC therapy.

## Figures and Tables

**Figure 1 polymers-16-02498-f001:**
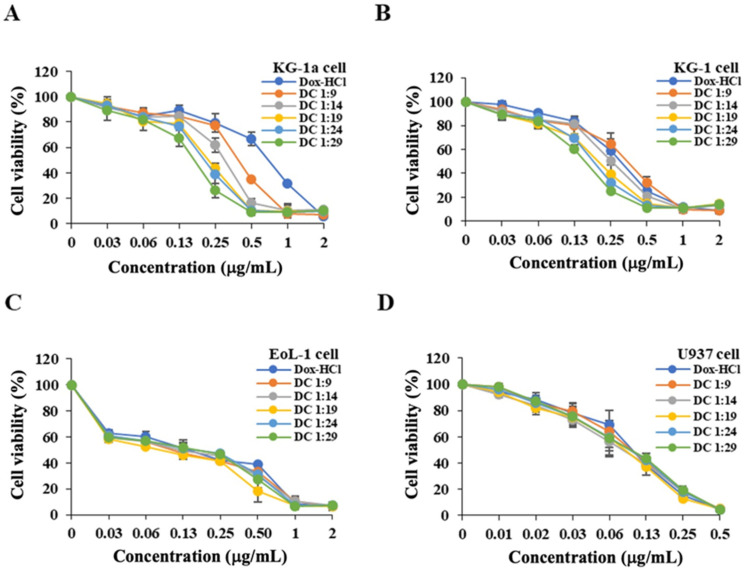
Cytotoxicity of Dox and Dox-Cur solution at various concentration ratios (1:9, 1:14, 1:19, 1:24, and 1:29) on AML leukemic stem cells and leukemic cells, including (**A**) KG-1a, (**B**) KG-1, (**C**) EoL-1, and (**D**) U937 cells. The data are shown as mean ± SD from 3 independent experiments.

**Figure 2 polymers-16-02498-f002:**
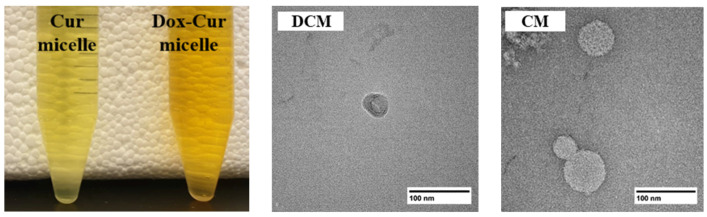
Physical appearance and morphology of CM and DCM from TEM.

**Figure 3 polymers-16-02498-f003:**
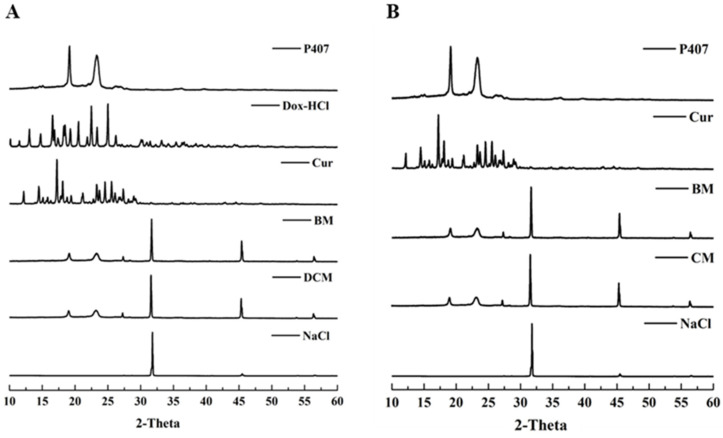
Crystalline characteristics of (**A**) DCM and (**B**) CM, compared to P407, Dox-HCl, Cur, and BM in powder form.

**Figure 4 polymers-16-02498-f004:**
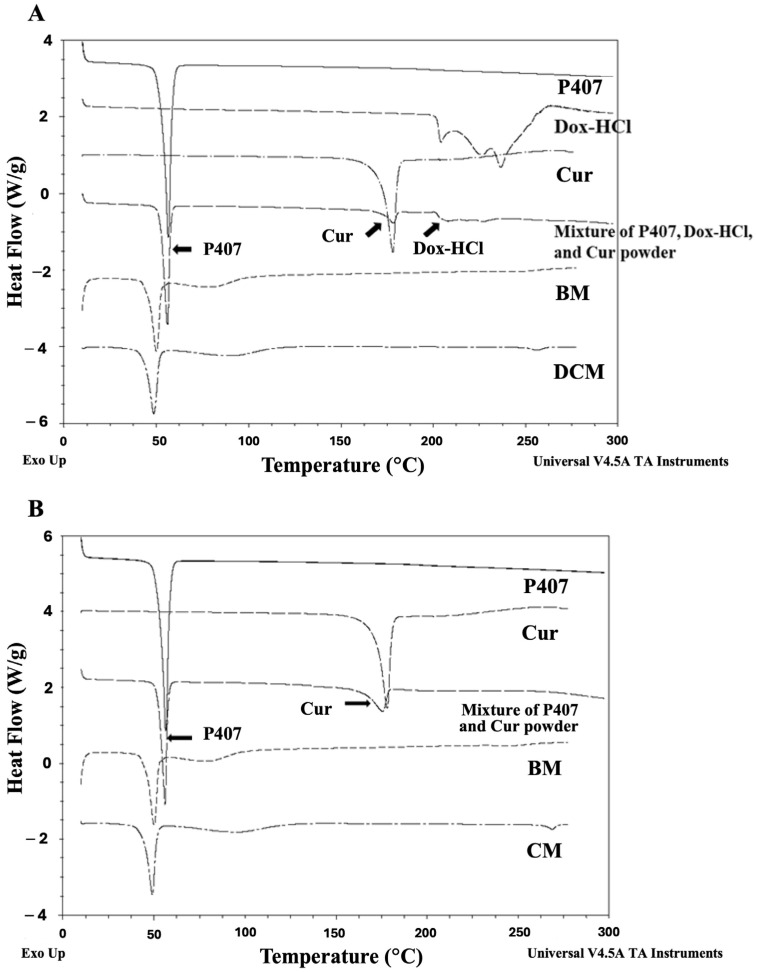
Melting point patterns of (**A**) DCM and (**B**) CM, compared to P407, Dox, Cur, drug mixture, and blank micelle (BM) in powder form.

**Figure 5 polymers-16-02498-f005:**
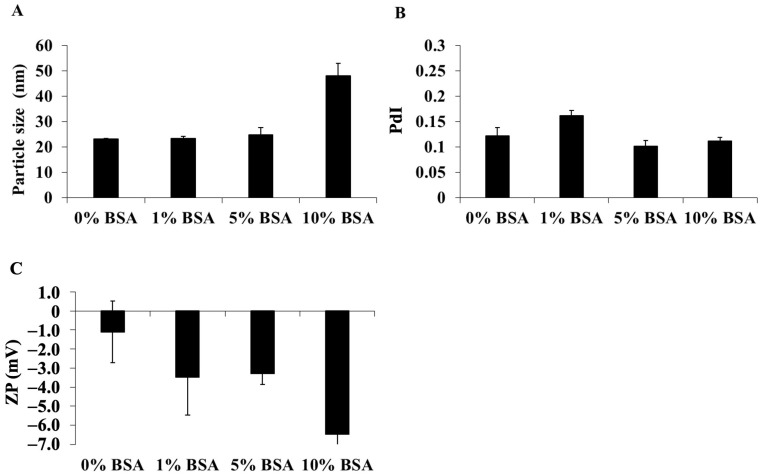
The alterations in (**A**) particle size, (**B**) polydispersity index (PdI), and (**C**) zeta potential value of DCM after 24 h of incubation in PBS, pH 7.4, containing several concentrations of BSA at 37 °C. The data are shown as mean ± SE from 3 independent experiments.

**Figure 6 polymers-16-02498-f006:**
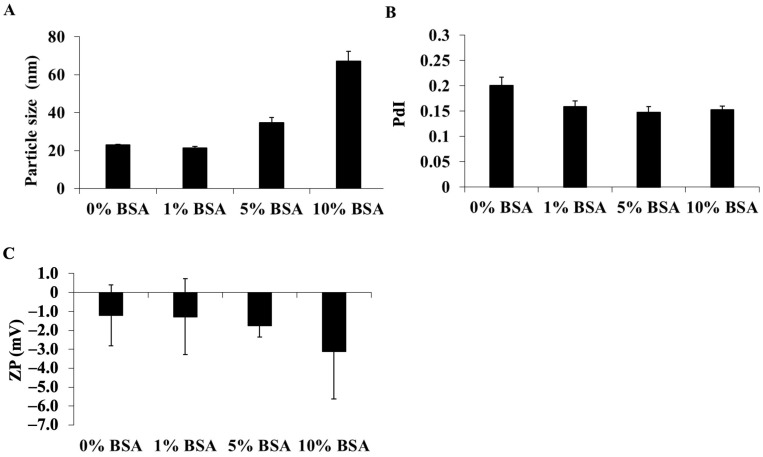
The alterations in (**A**) particle size, (**B**) polydispersity index (PdI), and (**C**) zeta potential value of CM after 24 h of incubation in PBS, pH 7.4, containing several concentrations of BSA at 37 °C. The data are shown as mean ± SE from 3 independent experiments.

**Figure 7 polymers-16-02498-f007:**
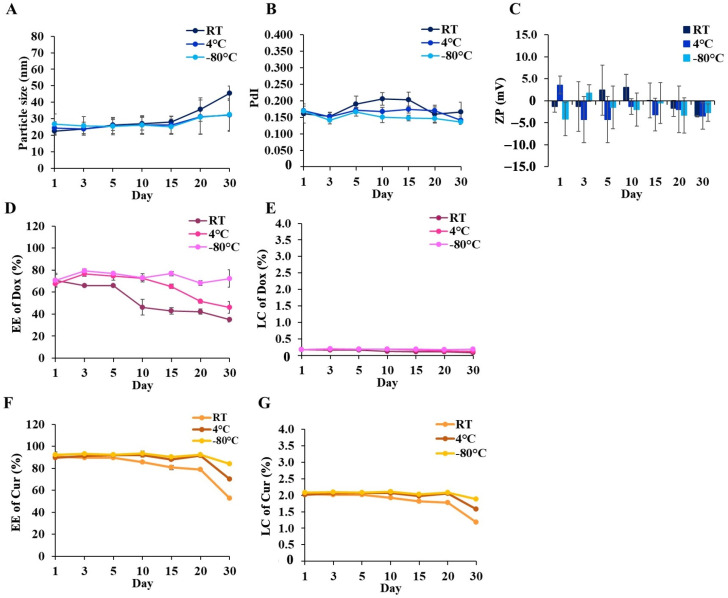
The variations in (**A**) particle size, (**B**) polydispersity index (PdI), (**C**) Zeta potential, (**D**) %EE of Dox, (**E**) %LC of Dox, (**F**) %EE of Cur, and (**G**) %LC of Cur in DCM after 30 days of storage at RT, 4 °C, and −80 °C in the dark. The data are shown as mean ± SE from 3 independent experiments.

**Figure 8 polymers-16-02498-f008:**
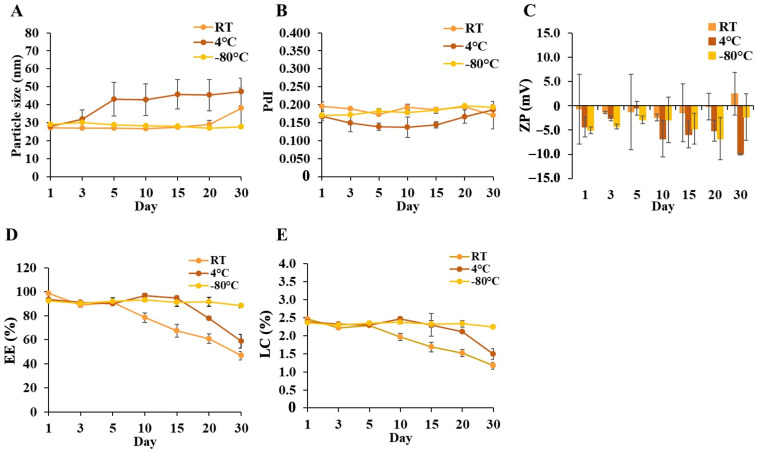
The variations in (**A**) particle size, (**B**) polydispersity index (PdI), (**C**) Zeta potential, (**D**) %EE, and (**E**) %LC of Cur in CM after 30 days of storage at room temperature (RT), 4 °C, and −80 °C in the dark. The data are shown as mean ± SE from 3 independent experiments.

**Figure 9 polymers-16-02498-f009:**
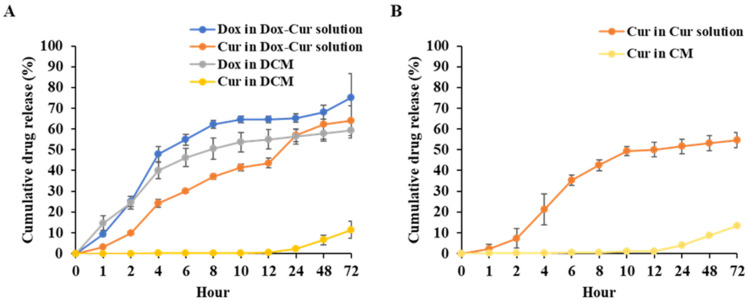
Drug release analysis of (**A**) Dox and Cur from DCM and Dox-Cur solution (**A**,**B**) Cur form CM and Cur solution in PBS, pH 7.4. The data are shown as mean ± SE from 3 independent experiments.

**Figure 10 polymers-16-02498-f010:**
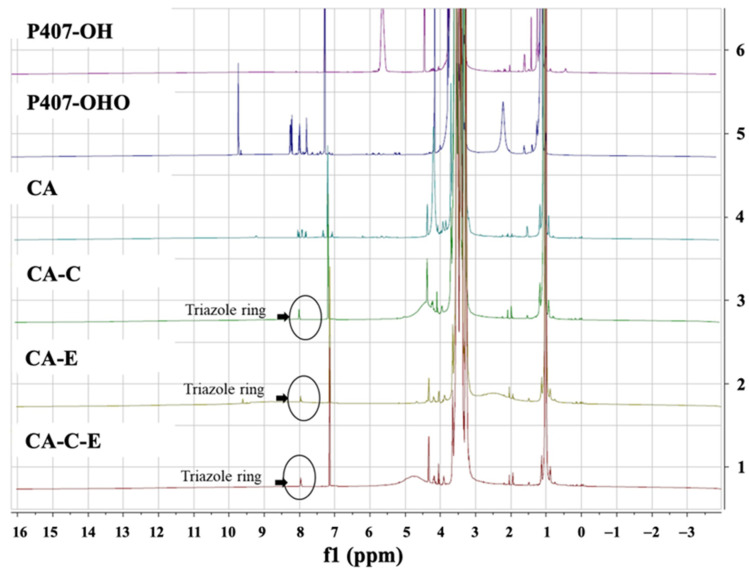
^1^H-NMR spectra of CA, CA-C, CA-E, and CA-C-E.

**Figure 11 polymers-16-02498-f011:**
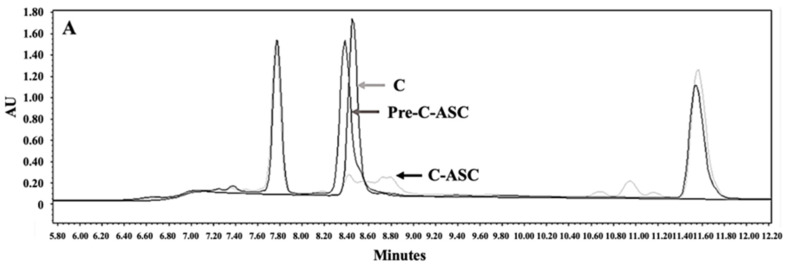
HPLC peaks demonstrated the conjugation of (**A**) CA-C, (**B**) CA-E, and (**C**) CA-C-E.

**Figure 12 polymers-16-02498-f012:**
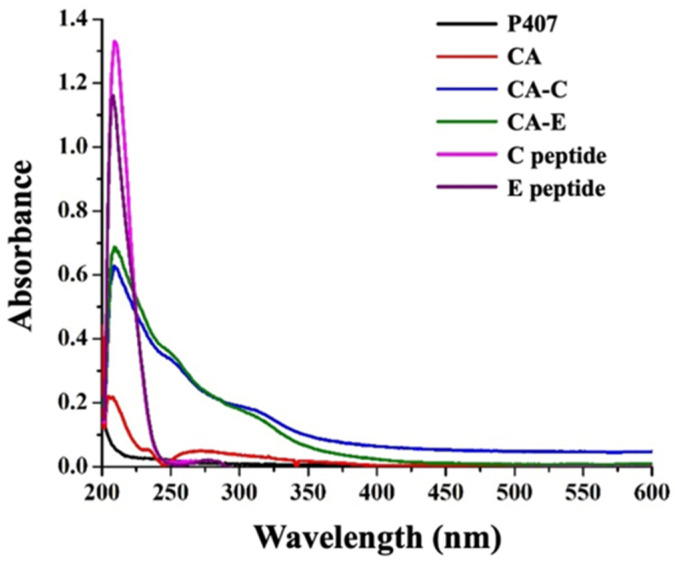
Characterization of cargo azide-conjugated FLT3 peptides using UV-Vis spectra of P407, CA, CA-C, CA-E, C peptide, and E peptide.

**Figure 13 polymers-16-02498-f013:**
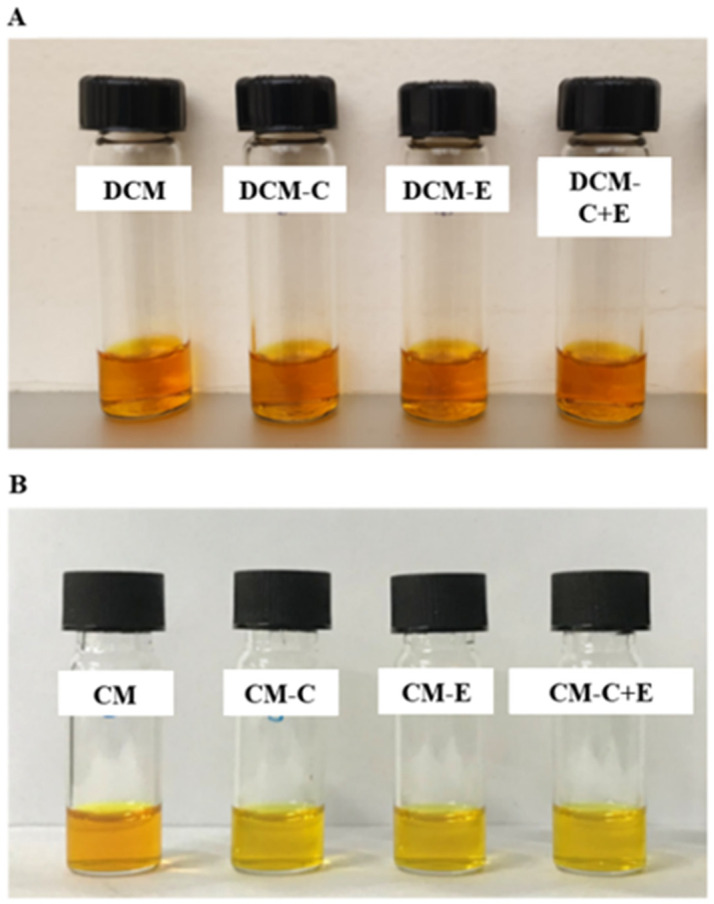
Physical appearance of (**A**) DCM, DCM-C, DCM-E, and DCM-C + E and (**B**) CM, CM-C, CM-E, and CM-C + E.

**Figure 14 polymers-16-02498-f014:**
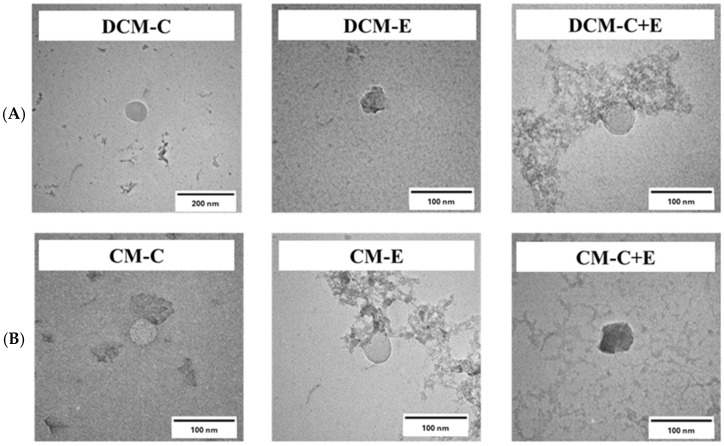
TEM images of (**A**) DCM-C, DCM-E, and DCM-C + E and (**B**) CM-C, CM-E, and CM-C + E.

**Figure 15 polymers-16-02498-f015:**
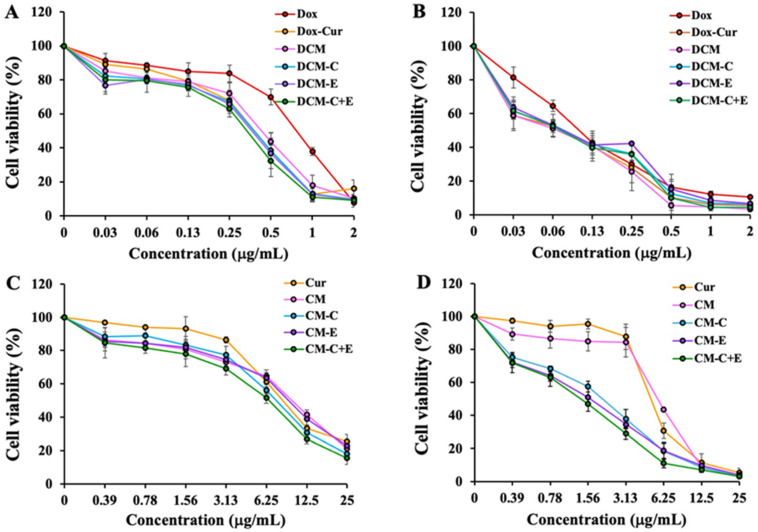
Cytotoxic effects of Dox solution, Dox-Cur solution, DCM, DCM-C, DCM-E, and DCM-C + DCM-E on (**A**) KG-1a and (**B**) EoL-1 cells. Cytotoxic effects of Cur solution, CM, CM-C, CM-E, and CM-C + CM-E on (**C**) KG-1a and (**D**) EoL-1 cells. The data are shown as mean ± SE from 3 independent experiments.

**Table 1 polymers-16-02498-t001:** IC_50_ values of Dox-HCl, Dox-Cur solution, DCM, DCM-C, DCM-E, and DCM-C + DCM-E in KG-1a and EoL-1 cells (mean ± SD, n = 3).

Sample	KG-1a Cells		EoL-1 Cells	
	IC_50_ of Dox (μg/mL)	IC_50_ of Cur (µg/mL)	IC_50_ of Dox (µg/mL)	IC_50_ of Cur (µg/mL)
Dox	0.81 ± 0.05	-	0.10 ± 0.00	-
Dox-Cur	0.40 ± 0.03 ^a^	3.61 ± 0.25	0.07 ± 0.01	0.67 ± 0.05
DCM	0.45 ± 0.03 ^a^	4.70 ± 0.28	0.06 ± 0.02	0.65 ± 0.18
DCM-C	0.41 ± 0.09 ^a^	3.38 ± 0.74	0.08 ± 0.01	0.66 ± 0.11
DCM-E	0.39 ± 0.06 ^a^	2.80 ± 0.44	0.07 ± 0.02	0.53 ± 0.15
DCM-C + E	0.36 ± 0.05 ^a^	2.71 ± 0.35	0.07 ± 0.01 ^a^	0.53 ± 0.08

^a^ *p* < 0.05 compared to Dox solution, n is number of independent experiments.

**Table 2 polymers-16-02498-t002:** IC_50_ values of Cur solution, CM, CM-C, CM-E, and CM-C + CM-E in KG-1a and EoL-1 cells (mean ± SD, n = 3).

Sample	IC_50_ (µg/mL)	
	KG-1a Cells	EoL-1 Cells
Cur	8.78 ± 0.40 ^b,e^	5.18 ± 0.31 ^b,c,d,e^
CM	10.19 ± 0.38 ^a,c,e^	5.75 ± 0.13 ^a,c,d,e^
CM-C	7.74 ± 0.52 ^b,d^	2.19 ± 0.30 ^a,b,d,e^
CM-E	9.68 ± 1.29 ^c,e^	1.55 ± 0.44 ^a,b,c^
CM-C + E	6.69 ± 0.75 ^a,b,d^	1.43 ± 0.27 ^a,b,c^

^a^ *p* < 0.05 compared to Cur solution, ^b^ *p* < 0.05 compared to CM, ^c^ *p* < 0.05 compared to CM-C, ^d^ *p* < 0.05 compared to CM-E, and ^e^ *p* < 0.05 compared to CM-C + CM-E.

## Data Availability

The original contributions presented in the study are included in the article/Appendix A, further inquiries can be directed to the corresponding authors.

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
