# Peer review of "Cytotoxicity of Doxorubicin-Curcumin Nanoparticles Conjugated with Two Different Peptides (CKR and EVQ) against FLT3 Protein in Leukemic Stem Cells"

_polymers, 2024, doi:10.3390/polym16172498_

Round 1

Reviewer 1 Report

Comments and Suggestions for Authors

The manuscript titled Cytotoxicity of doxorubicin-curcumin nanoparticle conjugated with two different peptides (CKR and EVQ) against FLT3 protein in leukemic stem cells as submitted to Polymers with manuscript Number Polymers-3164205 for possible publication is not suitable for the publication as such due to following points. Minor Revision should be asked.

1.      In Figure 1, graph A, B and C should have x-axis labelled as concentration not concentration of DOX as x-axis is showing concentration of both drugs.

2.      Isn’t zeta potential too less (-10 to +10mV)? It will lead to aggregation of micelles, even your studies show that micelles aggregate. Discuss please

3.      Verify the legends of Figure 2 and make it clear.

4.      Second and third paragraph in discussion seem aimless and need modification

5.      Discussion needs to be revise and improved

6.      Conclusion also need to be rewritten focusing the results obtained during the research work presented here.

7.      Tima et al., 2016 in line 946 and 992 when mentioning certain studies you can start with: last name et al. studied/prepared. No need to add year.

8.      References should be updated..Seems no reference from 2023 and 2024.

Comments on the Quality of English Language

Once authors should read the manuscript to remove minor mistakes.

Author Response

Comment 1. In Figure 1, graph A, B and C should have x-axis labelled as concentration not concentration of DOX as x-axis is showing concentration of both drugs.

Response 1: “Concentration of DOX” has been changed to “Concentration” already on page 8.

Comment 2. Isn’t zeta potential too less (-10 to +10mV)? It will lead to aggregation of micelles, even your studies show that micelles aggregate. Discuss please

Response 2: This point has been discussed in discussion part already on page 21, lines 960-964 with yellow highlight.

“Numerous micelle reports [27,30,47,48] have indicated a low zeta potential can cause particle agglomeration due to van der Waals forces [16]; however, the hydrophilic PEO block of P407 has the capability to prevent aggregation, protein absorption, and recognition of the particle by RES [25,47].”

Comment 3. Verify the legends of Figure 2 and make it clear.

Response 3: Figure 2 has been verified the legend already. It has been changed fromPhysical appearance and morphology of Blank micelles (BM), Dox-micelles (DM), Cur-micelles (CM), and Dox-Cur-micelles (DCM) from TEM.”  to “Physical appearance and morphology of CM and DCM from TEM.” on page 9, lines 406-407.

Comment 4. Second and third paragraph in discussion seem aimless and need modification

Response 4: Second and third paragraph in discussion have been removed as shown on pages 20-21.

Comment 5. Discussion needs to be revise and improved

Response 5: Discussion have been revised and improved with track changes on pages 20-23.

Comment 6. Conclusion also need to be rewritten focusing the results obtained during the research work presented here.

Response 6: Conclusion has been revised with track changes on pages 23-24.

Comment 7. Tima et al., 2016 in line 946 and 992 when mentioning certain studies you can start with: last name et al. studied/prepared. No need to add year.

Response 7: Year after Tima et al. was deleted already.

Comment 8. References should be updated..Seems no reference from 2023 and 2024.

Response 8: References have been updated. References numbers 37, 38, and 39 was published in years 2023 and 2024.

References

  1. Liu, Y.; Liu, Y.; Sun, X.; Wang, Y.; Du, C.; Bai, J. Morphologically transformable peptide nanocarriers coloaded with doxorubicin and curcumin inhibit the growth and metastasis of hepatocellular carcinoma. Materials Today Bio 2024, 24, 100903.
  2. Tang, H.; Li, L.; Wang, B.; Medicine, G.S.R.C.o.T.C. Observation of antitumor mechanism of GE11-modified paclitaxel and curcumin liposomes based on cellular morphology changes. AAPS Open 2024, 10, 1.
  3. Chen, M.; Fang, X.; Du, R.; Meng, J.; Liu, J.; Liu, M.; Yang, Y.; Wang, C. A Nucleus-Targeting WT1 Antagonistic Peptide Encapsulated in Polymeric Nanomicelles Combats Refractory Chronic Myeloid Leukemia. Pharmaceutics 2023, 15, 2305.

Reviewer 2 Report

Comments and Suggestions for Authors

Title: Cytotoxicity of Doxorubicin-Curcumin Nanoparticle Conjugated with Two Different Peptides (CKR and EVQ) Against FLT3 Protein in Leukemic Stem Cells

Manuscript ID: polymers-3164205

I have carefully reviewed the manuscript titled "Cytotoxicity of Doxorubicin-Curcumin Nanoparticle Conjugated with Two Different Peptides (CKR and EVQ) Against FLT3 Protein in Leukemic Stem Cells," referenced as polymers-3164205, which was submitted to MDPI Polymers.

This study focuses on developing a targeted micellar delivery system combining Doxorubicin (Dox) and Curcumin (Cur) to improve the efficacy and reduce the toxicity of these drugs in treating leukemic stem cells (LSCs), specifically in KG1a cells. By formulating Dox-Cur-micelles (DCM) and Cur-micelles (CM) with the addition of FLT3-specific peptides, the researchers aim to enhance drug delivery and increase cytotoxicity in LSCs compared to other leukemic cells like EoL-1. The micelles were prepared using a film-hydration technique combined with pH-induced self-assembly, achieving stable formulations with favorable particle size and drug encapsulation efficiency. The results indicate that micelles conjugated with two FLT3-targeting peptides exhibited greater toxicity in both KG-1a and EoL-1 cells, suggesting an improved likelihood of effective drug binding and enhanced therapeutic potential against leukemia.

Based on my review, I recommend the manuscript for publication with minor revisions. The research presents a promising and innovative approach to enhancing the cytotoxicity of Doxorubicin and Curcumin nanoparticles in leukemic stem cells, with clear potential for clinical application. However, I suggest addressing the following points before publication:

-Simplify and clarify some of the technical language in the abstract and main text to make the study more accessible to a broader audience.

- Enhance the discussion of the results, particularly in relation to existing treatments, and emphasize the potential clinical implications of the findings.

-Include a discussion of recent studies that relate to your work to better position your findings within the broader context of leukemia treatment research.

-The topic is innovative and very pertinent for the area of focused cancer treatment. Examining the dual-targeting of FLT3 receptors in leukemic stem cells closes a significant gap in the evolution of nanoparticle-based medication delivery systems. This study is relevant given the continuous difficulties in efficiently treating leukemic stem cells and might help to improve tailored treatments, which are much needed in the area.
Contribution to the Domain:

-By proving a novel approach to improving the cytotoxic effects of Doxorubicin and Curcumin by the conjugation of two particular peptides (CKR and EVQ), this work adds value to the current literature. This dual-targeting technique may provide a more efficient way to eradicate leukemic stem cells, therefore lowering the chance of recurrence, unlike much published research that concentrates on single-target tactics.

-The results line up with the reasoning and data the book offers. The primary research issue is mainly addressed by the work, which shows that dual-targeting FLT3 receptors with conjugated nanoparticles increases cytotoxicity in leukemic stem cells. The findings confirm the theory and the logical inferences reached from the facts support each other.
References:

-The sources mentioned in the book are pertinent and fit for the issue. More recent research into comparable nanoparticle-based treatments or dual-targeting strategies in cancer therapy might be advantageous, nevertheless.
These comments should assist in improving the work and guarantee that it significantly advances the area.

Comments:

Please write about the novelty of the research

In the figure, Please add the close bracket in “%”

Please change Figure 4, with high-resolution

To improve the clarity and focus of your manuscript, I recommend removing the 24-hour time point from Figures 5 and 6, as it appears to be unnecessary

By addressing these minor revisions, the manuscript will be better positioned to make a significant contribution to the field.

Comments on the Quality of English Language

Minor editing of the English language is required.

Author Response

Comment 1: Simplify and clarify some of the technical language in the abstract and main text to make the study more accessible to a broader audience.

Response 1: Some of the technical terms in the abstract and main text have been added for more understanding using track changes on page 1, lines 27, 28, and 32.

Comment 2: Enhance the discussion of the results, particularly in relation to existing treatments, and emphasize the potential clinical implications of the findings.

 Response 2: We added the sentence “From this study, the co-delivery of drugs in polymeric micelle conjugated with two peptides benefits to improve the potential clinical implications for AML patients in the future.” In the discussion to enhance the discussion of the results on page 23, lines 1053-1055.

Comment 3: Include a discussion of recent studies that relate to your work to better position your findings within the broader context of leukemia treatment research.

Response 3: Thank you very much for your suggestions. We also added recently reports in the manuscript. The recent studies in 2022-2024 were found in breast cancer, lung cancer, and hepatocarcinoma page 21, lines 917-931. There have been limited studies focused on developing the delivery of Dox and Cur using nanoparticles conjugated with cell specific peptides, particularly in AML.

Comment 4: The topic is innovative and very pertinent for the area of focused cancer treatment. Examining the dual-targeting of FLT3 receptors in leukemic stem cells closes a significant gap in the evolution of nanoparticle-based medication delivery systems. This study is relevant given the continuous difficulties in efficiently treating leukemic stem cells and might help to improve tailored treatments, which are much needed in the area.
Contribution to the Domain:
Response 4: Thank you very much for your valuable comments. We also hope this recent study could contribute to study in nanotechnology and clinical application domains in the future. However, this suggestion has been added to the conclusion part on page 24, lines 1083-1085.

Comment 5: By proving a novel approach to improving the cytotoxic effects of Doxorubicin and Curcumin by the conjugation of two particular peptides (CKR and EVQ), this work adds value to the current literature. This dual-targeting technique may provide a more efficient way to eradicate leukemic stem cells, therefore lowering the chance of recurrence, unlike much published research that concentrates on single-target tactics.

Response 5: Thank you very much for a good comment. Our research is a novel strategy to improve drug delivery system against cancer cell targeting. Combinations of drug and curcumin as well as two particular peptides (CKR and EVQ) have been designed to provide an efficient way to eradicate leukemic stem cells. There is no previous report of this strategy before. However, our group used to publish Dox micelles conjugated with only one peptide (CKR or EVQ) in the Journal also (Chueahongthong et al., 2022). This report studied the production of Dox-micelle conjugated FLT3 peptides, which was proven to be effective in increasing Dox uptake and inducing apoptosis in LSC-like KG-1a cells.

Reference

Chueahongthong, F.; Tima, S.; Chiampanichayakul, S.; Dejkriengkraikul, P.; Okonogi, S.; Sasarom, M.; Rodwattanagul, S.; Berkland, C.; Anuchapreeda, S. Doxorubicin-Loaded Polymeric Micelles Conjugated with CKR- and EVQ-FLT3 Peptides for Cytotoxicity in Leukemic Stem Cells. Pharmaceutics 2022, 14. 10.3390/pharmaceutics14102115.

Comment 6: The results line up with the reasoning and data the book offers. The primary research issue is mainly addressed by the work, which shows that dual-targeting FLT3 receptors with conjugated nanoparticles increase cytotoxicity in leukemic stem cells. The findings confirm the theory, and the logical inferences reached from the facts support each other.
References:
Response 6: According to our previous report (Chueahongthong et al., 2022), Dox nanoparticles with two peptides (CKR and EVQ) which were DM-CKR + DM-EVQ showed the good cytotoxicity against leukemic stem cells. Thus, this study, we demonstrated the novel model of co-delivery of Dox and Cur in polymeric micelle conjugated with two different peptides for drug delivery system improvement. The binding of both peptides to the FLT3 protein receptor close to the high-affinity binding region associated with domains 2 and 3 of the FLT3 extracellular domain. The finding supported that the use of Dox and Cur micelles with dual-targeting FLT3 receptors with conjugated nanoparticles increase cytotoxicity in leukemic stem cells.

Reference

Chueahongthong, F.; Tima, S.; Chiampanichayakul, S.; Dejkriengkraikul, P.; Okonogi, S.; Sasarom, M.; Rodwattanagul, S.; Berkland, C.; Anuchapreeda, S. Doxorubicin-Loaded Polymeric Micelles Conjugated with CKR- and EVQ-FLT3 Peptides for Cytotoxicity in Leukemic Stem Cells. Pharmaceutics 2022, 14. 10.3390/pharmaceutics14102115.

Comment 7: The sources mentioned in the book are pertinent and fit for the issue. More recent research into comparable nanoparticle-based treatments or dual-targeting strategies in cancer therapy might be advantageous, nevertheless. These comments should assist in improving the work and guarantee that it significantly advances the area.

 Response 7: Thank you very much for your suggestions. We also added recently reports in the manuscript.

Comment 8: Please write about the novelty of the research

Response 8: The novelty of research has been added in the introduction part on page 3, lines 125-126 “This study demonstrates the novel model of nanoparticle conjugated with two different peptides for drug delivery system improvement.”. Moreover, in the discussion part, we also added the sentences “This is the first report to design a novel strategy model for drug delivery system.” on page 20, line 943-944 to support the novelty of the research. However, we claimed that the technique in this study was successful to produce a new model of micelle containing two peptides. Furthermore, co-delivery Dox and Cur loaded micelles that developed using P407 polymer and two peptides had never been reported before.

Comment 9: In the figure, Please add the close bracket in “%”

Response 9: The close brackets were added in “%” already in Figures 1, 7, 8, and 15.

Comment 10: Please change Figure 4, with high-resolution

Response 10: Figure 4 was edited for high-resolution already on pages 10-11.

Comment 11: To improve the clarity and focus of your manuscript, I recommend removing the 24-hour time point from Figures 5 and 6, as it appears to be unnecessary

 Response 11: “24-hour” has been removed from Figures 5 and 6 already on pages 11 and 12.

Comment 12: By addressing these minor revisions, the manuscript will be better positioned to make a significant contribution to the field.

Response 12: Thank you very much for your valuable suggestions.

Comment 13: Minor editing of the English language is required.

Response 13: English language has been edited by native speaker (Prof.Dr. Cory Berkland) to improve the quality as shown in track changes.

Reviewer 3 Report

Comments and Suggestions for Authors

improve the he resolution of figures.

Comments on the Quality of English Language

I would like to thank you for choosing me as a reviewer in your journal.

The manuscript “Cytotoxicity of Doxorubicin-Curcumin Nanoparticle Conjugated with Two Different Peptides (CKR and EVQ) Against FLT3 Protein in Leukemic Stem Cells” by Chueahongthong  et al. reports an original good work about Dox-Cur-micelles (DCM) were formulated and increase specificity to 120 targeted cancerous cells by conjugation with two synthetic FLT3-specific peptides (CKR; 121 C and EVQ; E) on the surface of DCM (DCM-C+E) were developed to improve the efficacy 122 of Dox and Cur delivery to FLT3-positive AML cell lines, especially AML-LSCs. The manuscript is fully suited to publication in Polymer Journal due to the originality and novelty of the topic covered. 

Improve the English Language in the manuscript. 

Author Response

Comment 1: improve the resolution of figures.

Response 1: Almost figures have been improved the resolution already.

Comment 2: Comments on the Quality of English Language

Response 2: English language has been edited by native speaker (Prof.Dr. Cory Berkland) to improve the quality as shown in track changes.